# Development of Machine Learning Models for Prediction of Osteoporosis from Clinical Health Examination Data

**DOI:** 10.3390/ijerph18147635

**Published:** 2021-07-18

**Authors:** Wen-Yu Ou Yang, Cheng-Chien Lai, Meng-Ting Tsou, Lee-Ching Hwang

**Affiliations:** 1Department of Neurology, Taipei Veterans General Hospital, Taipei City 11217, Taiwan; urania41228@gmail.com; 2Department of Medicine, Taipei Veterans General Hospital, Taipei City 11217, Taiwan; kelvin2005s@gmail.com; 3Department of Family Medicine, Mackay Memorial Hospital, Taipei City 10491, Taiwan; mttsou@gmail.com; 4Mackay Junior College of Medicine, Nursing and Management, Taipei City 11260, Taiwan; 5Department of Medicine, Mackay Medical College, New Taipei City 252, Taiwan

**Keywords:** osteoporosis, prediction model, machine learning, screening tool, early detection

## Abstract

Osteoporosis is treatable but often overlooked in clinical practice. We aimed to construct prediction models with machine learning algorithms to serve as screening tools for osteoporosis in adults over fifty years old. Additionally, we also compared the performance of newly developed models with traditional prediction models. Data were acquired from community-dwelling participants enrolled in health checkup programs at a medical center in Taiwan. A total of 3053 men and 2929 women were included. Models were constructed for men and women separately with artificial neural network (ANN), support vector machine (SVM), random forest (RF), k-nearest neighbor (KNN), and logistic regression (LoR) to predict the presence of osteoporosis. Area under receiver operating characteristic curve (AUROC) was used to compare the performance of the models. We achieved AUROC of 0.837, 0.840, 0.843, 0.821, 0.827 in men, and 0.781, 0.807, 0.811, 0.767, 0.772 in women, for ANN, SVM, RF, KNN, and LoR models, respectively. The ANN, SVM, RF, and LoR models in men, and the ANN, SVM, and RF models in women performed significantly better than the traditional Osteoporosis Self-Assessment Tool for Asians (OSTA) model. We have demonstrated that machine learning algorithms improve the performance of screening for osteoporosis. By incorporating the models in clinical practice, patients could potentially benefit from earlier diagnosis and treatment of osteoporosis.

## 1. Introduction

Osteoporosis is characterized by decreased bone density and architectural disruption of the bone tissue, leading to a susceptibility to fractures [1,2]. Osteoporosis may cause disability and mortality in individuals [3], and is recognized as an important public health problem worldwide [4]. People diagnosed with osteoporosis is strikingly increasing and facing the greatest challenge to prevalence issues owing to the rapidly aging society [4,5,6]. The prevalence of osteoporosis and osteoporotic fractures among Taiwanese more than 50 years old increased from 17.4% in 2001 to 25.0% in 2011 [1]. Treatment modalities with lifestyle change, fall precaution, and medication have been proposed and results in a 21–66% reduction in fracture risks in osteoporosis patients [7]. Hence, early intervention to prevent fractures can be achieved through earlier osteoporosis detection. 

With dual-energy *X*-ray absorptiometry (DXA) being an important tool for diagnosis [8,9], the availability of DXA is quite limited and the utility of DXA for osteoporosis diagnosis is extremely low, with about 0.95% annually in adults above 50 years old [5]. Currently, bone mineral density (BMD) examination is still not included as a generalized screening tool during regular health checkup in Taiwan. Moreover, the presence of osteoporosis is often overlooked in clinical practice [10]. To identify patients at risk and to increase the awareness of physicians for asymptomatic osteoporosis, understanding the risk factors and proper interpretation is crucial.

Osteoporosis is associated with age, female gender, low body weight, low physical activity, poor nutritional status, and other endocrine and cardiometabolic factors [11,12,13,14,15,16,17,18,19]. Beyond the identification of risk factors, many researchers also aimed to develop prediction models for the screening of osteoporosis. The Osteoporosis Self-Assessment Tool for Asians (OSTA) model was developed for postmenopausal Asian women [20]. Another study constructed a logistic regression model [21]. Aside from traditional modeling, machine learning algorithms are gaining popularity in recent years for their flexible modelling and the ability to detect more complex relationship between input features and outputs, thus enhancing the performance of prediction [22,23]. In the field of osteoporosis, several studies aimed to predict fracture risks in osteoporosis patients [24], and some aimed to predict BMD with *X*-ray or computed tomography images [25]. For prediction of osteoporosis with more easily available data, some machine learning models were constructed, focusing mainly on postmenopausal women [26,27]. These models performed significantly better comparing with traditional ones. Currently, there is still a lack of machine learning approach for prediction of osteoporosis with a larger dataset in both men and women.

In our study, we aimed to develop prediction models with machine learning algorithms for screening of osteoporosis in community-dwelling individuals more than fifty years old, using easily available input features consisted of physical characteristics, personal and medical history, and laboratory tests. By incorporating the prediction model in clinical practice as a screening tool, both the patients and the physicians could be more aware of the risk of osteoporosis, and take further action in the early stage to prevent undesirable outcomes.

## 2. Materials and Methods

### 2.1. Data Acquirement

Data of community-dwelling individuals who participated in health checkup programs between 2008 and 2018 in a medical center in northern Taiwan were reviewed. In all participants, past medical history and personal history were taken, physical examination were performed including measurement of vital signs, body height, body weight, and waist circumference. Hematological and Biochemical profiles were obtained but the specific tested items differed in individual program. BMD was checked with DXA (Lunar Prodigy Advance; GE Healthcare, Madison, WI, USA) at the lumbar spine and bilateral hip joint. *T*-score represented the standard deviation of BMD from the healthy young adults of same sex and ethnicity. The BMD results were compared with the results from healthy young adults. The *T*-scores at each site were obtained, and the lowest *T*-score was used to interpret the results as osteoporosis (*T*-score ≤ −2.5), osteopenia (−2.5 < *T*-score < −1), or normal (*T*-score ≥ −1) in the final report card. This study was approved by the Ethics Committee of Mackay Memorial Hospital (Institutional review board number: 18MMHIS137).

Out of 18,629 enrollments, 12,647 were excluded due to DXA not performed, age under 50 years old, repeated enrollment of the same individual, extreme outliers including creatinine > 10 mg/dL, Hemoglobin A1c (HbA1c) > 15%, alanine transaminase (ALT) > 1000 IU/L, and other missing data. The remaining 5982 enrollments were included for further analysis. The flowchart of data inclusion and preprocessing is shown in Figure 1.

### 2.2. Feature Selection and Data Preprocessing

Candidate features were categorized into different domains related to bone health, including physical characteristics, history of smoking, history of alcohol drinking, medical history of diabetes mellitus, medical history of hypertension, hematological profile, renal function, liver function, thyroid function, lipid profile, protein content in blood, electrolytes, and for women, obstetrics and gynecological history. Features from these domains were examined for the representativeness, availability, and the significance on the prediction of decreased bone density. We selected 1–2 features from each domain which were more representative, more commonly used, and showed significant difference between the normal BMD group and the decreased BMD group. This selection process was done by discussion among the authors. After examination, 16 input features for men and 19 input features for women were selected. A summary of candidate and selected features was presented in Appendix A.

The input features applied to the models included: age, body weight, body height, waist circumference, history of smoking, history of alcohol drinking, diabetes mellitus (DM), hypertension (HTN), and blood test results of albumin, hemoglobin, ALT, creatinine, triglyceride (TG), high-density lipoprotein cholesterol (HDL-C), alkaline phosphatase (ALK-P), and thyroid-stimulating hormone (TSH). For women, the menopause status, history of hormone-replacement therapy (HRT), and parity were also included. Alcohol drinking was defined as drinking more than once per week, diabetes mellitus was defined as having a previous diagnosis, fasting plasma sugar level ≥ 126 mg/dL, postprandial plasma sugar level ≥ 200 mg/dL, or HbA1c level ≥ 6.5%, and hypertension was defined as having a previous diagnosis, systolic blood pressure (SBP) ≥ 140 mmHg, or diastolic blood pressure (DBP) ≥ 90 mmHg. 

The aim of this study was to predict the presence of osteoporosis. However, the prevalence of osteoporosis was low in our study population (10.4% in women and 3.8% in men), and this imbalanced class proportion would cast obstacles to the machine learning process. To overcome the problem, and to preserve more information about the relationship between input features and low bone mass, we trained the model with prediction of “decreased bone density”, which consisted of the osteopenia and osteoporosis group. During the training process, the prediction target of the model was a binary variable (1, 0), where ‘1’ represented the decreased bone density groups, and ‘0’ represented the normal group. In contrast, during the testing process, the prediction results were tested with ‘1’ standing for the osteoporosis group, and ‘0’ standing for the normal and osteopenia group. 

Data in both men and women were randomly divided into training and testing dataset with an 80:20 split. This resulted in 2442 and 611 men in the training and testing datasets, and 2343 and 586 women in the training and testing datasets, respectively. For the artificial neural network (ANN), support vector machine (SVM), and k-nearest neighbor (KNN) models, the training dataset was rescaled to the range of 0 to 1, and the testing dataset was rescaled using the rescale index of the training dataset.

### 2.3. Machine Learning Model Development

Candidate machine learning algorithms in the study included ANN, SVM, random forest (RF), KNN, and logistic regression (LoR). The ANN model was constructed with Tensorflow 1.14.0 (Google Brain Team, Mountain View, CA, USA), and the other models with Scikit-learn 0.21.2 [28], under the environment of Python 3.7 (Python Software Foundation, Wilmington, DE, USA). Due to the difference in input features, baseline characteristics, and prevalence of positive prediction, the models were trained separately in men and women.

During each training section of the ANN, SVM, RF, and KNN model, a 20% validation dataset was randomly split out from the training dataset to test for the performance of the models. On the contrary, the full training dataset was used during construction of the LoR model. To avoid overfitting, we limited the complexity of the models, and applied early stopping and dropout during training of the ANN model [29]. For early stopping, we monitored the AUROC of the training and validation dataset in each epoch. Training was stopped when the AUROC of the validation dataset reached a peak and did not show further improvement after 100 epochs. The model generated from the peak point was used for further analysis.

For hyperparameter tuning, different combinations of the number of layers (1 or 2 hidden layers), number of nodes (from 4 to 20 in each hidden layer), learning rate (from 0.01 to 0.0001), and dropout rate (from 0% to 60%) were tested for the ANN model. Grid search was done for the SVM, RF, and KNN models. For the SVM model, the kernel type (linear, polynomial, or radial basis function), regularization parameter C (from 2^−2^ to 2^9^), kernel coefficient gamma (from 2^−9^ to 2^2^), and degree (from 1 to 4) for the polynomial kernel function were examined; for the RF model, the number of trees (from 100 to 1000), number of features to consider (from 3 to 11), and the maximum depth of the tree (from 3 to 11) were examined; for the KNN model, the number of neighbors (from 1 to 30), the leaf size (from 1 to 49), and the power parameter *p* (Manhattan distance or Euclidean distance) were tested. All models were constructed with a balanced class weight. For each hyperparameter set, randomization for the split of validation dataset and the training process was repeated several times. The area under the receiver operating characteristic curve (AUROC) was calculated for the validation dataset in each training process, and the mean value of AUROC were compared. We adopted the hyperparameter that yield a better mean AUROC during validation to train our models.

The final adopted hyperparameter for ANN model was with two hidden layers: 9 nodes in hidden layer one, 4 nodes in hidden layer two in men’s model, and 13 nodes in hidden layer one, 7 nodes in hidden layer two in women’s model. The learning rate was set to 0.001, optimized with adaptive moment estimation, and the dropout rate was set to 0.4 in both genders included in the study. For the SVM model, the better performance was obtained with radial basis function kernel, for men’s and women’s model, the regularization parameter C was set to 32 and 16, and kernel coefficient gamma was set to 0.125 and 0.5, respectively. In the RF model, the number of trees was set to 300, the number of features to consider was set to 8, and the maximal depth of tree was set to 8 in the men’s model; the number of trees was set to 600, the number of features to consider was set to 10, and maximal depth of the tree was set to 10 in the women’s model. Finally, for the KNN model, the number of neighbors was set to 28, the leaf size was set to 15, and the power parameter *p* was set to the Manhattan distance in the men’s model; the number of neighbors was set to 13, the leaf size was set to 3, and the power parameter, *p*, was set to the Euclidean distance in the women’s model. The process and the final selected hyperparameters for each model were presented in Table 1.

After we determined the hyperparameter in each model, the model was trained again with the specific set of hyperparameter several times to obtain the maximal AUROC of the validation dataset, and the final product was further analyzed with the testing dataset.

### 2.4. Statistical Analysis

We applied the testing dataset to different models to obtain the predicted probabilities of having osteoporosis in each model. When we tested these probabilities with the true condition set to having or not having osteoporosis, receiver operating characteristic (ROC) curves could be drawn. AUROC was calculated to examine and compare the performance of different machine learning models. A 95% confidence interval (CI) for the AUROC and comparison between different AUROC values were performed with MedCalc 19.2, using methods provided by DeLong et al. [30]. 

To compare the performance of the machine learning models in our study and traditional models, the OSTA score [20] was included. The OSTA score was calculated as 0.2(Weight (kg)–Age (year)), with the results rounded to an integer. We applied the OSTA score to the testing dataset, and the AUROC was also calculated and compared.

For cutoff point optimization, a weighted Youden index [31,32] was applied with a weight of 0.6. The weight was selected at 0.6 to increase the sensitivity, also not sacrificing too much on specificity, as we aimed to construct a screening tool for osteoporosis in the general population. The sensitivity and specificity were calculated at the corresponding cutoff point which maximized the weighted Youden index.

## 3. Results

### 3.1. Demographic Information of the Study Population

Of the 5982 participants enrolled in the study, 3053 (51.0%) were men and 2929 (49.0%) were women. The average age was 59.3 ± 7.0 years old for both gender groups. The results of DXA showed that 117 men (3.8% of men) and 304 women (10.4% of women) had osteoporosis, and 1134 men (37.1% of men) and 1369 women (46.7% of women) had osteopenia. Other demographic information was summarized in Table 2.

For input feature selection, we compared the candidate features in the normal bone density group and the decreased bone density group. Most of the selected features showed significance. Other features that did not show significance, including DM, HTN, and history of HRT, were still included for their well-established relation to bone health. The comparison of input features between participants with normal bone density and decreased bone density was done. The results were shown in Table 3.

### 3.2. Model Performance

In the five machine learning models (ANN, SVM, RF, KNN, LoR) and the traditional model (OSTA), the AUROC, sensitivity, and specificity were presented in Table 4. The sensitivity and specificity were calculated at the cutoff value determined by maximization of weighted Youden index at weighted 0.6. The resultant cutoff value was <3 in men and <0 for women in the OSTA model. Overall, for the prediction of osteoporosis in men, the machine learning models obtained 83–96% sensitivity and 53–73% specificity; for the prediction of osteoporosis in women, the machine learning models obtained 76–90% sensitivity and 62–69% specificity. Additionally, we compared the AUROC between the machine learning models and OSTA. The ANN, SVM, RF, LoR models in men, and the ANN, SVM, RF models in women performed significantly better than the OSTA model. The ROC curves of different machine learning models in men and women were presented in Figure 2A,B.

## 4. Discussion

In this study, we used five different machine learning algorithms as ANN, SVM, RF, KNN, and LoR for the screening of osteoporosis in community-dwelling individuals older than fifty years old. The models reached AUROC of 0.821–0.843 in men and 0.767–0.811 in women. At designated cutoff points, the models achieved 83–96% sensitivity and 53–73% specificity on prediction of osteoporosis in men; and 76–90% sensitivity and 62–69% specificity on prediction of osteoporosis in women. The ANN, SVM, RF, and LoR models in men, and the ANN, SVM, and RF models in women performed significantly better than the well-established OSTA model.

Previous works on this topic focused more on the women population. A study published in 2013 by Kim et al. including 1674 postmenopausal Korean women [27] gained the best AUROC at 0.827 with the SVM model for prediction of osteoporosis using age, height, weight, body mass index, duration of menopause, duration of breast feeding, estrogen therapy, hyperlipidemia, hypertension, osteoarthritis, and diabetes mellitus as input features. The model reported 77.8% sensitivity and 76.0% specificity. In another study published in 2020 by Shim et al. [26], also targeting 1792 postmenopausal women and compared seven machine learning models. The best performance was achieved with the ANN model, with an AUROC of 0.743. One research published in 2019 by Meng et al. included study population of women older than twenty years old [33]. An ANN model was constructed and the AUROC reached 0.829. Comparing with these studies, we are the first to include men in the prediction of osteoporosis. Among male patients, secondary etiology accounts for up to 65% of cases in osteoporosis; these secondary causes include alcohol abuse, steroid treatment and other metabolic disorders. In contrast, the prevalence of secondary osteoporosis in women is much lower as compared to men. Postmenopausal estrogen deficiency and senile osteoporosis stands for the main causes of primary osteoporosis in women [34]. Due to the difference in etiology and baseline characteristics, we trained prediction models for men and women separately, and gained satisfactory results. The advantages in our study included the larger dataset in both men (n = 3053) and women (n = 2929), and the inclusion of more input features (16 features in men and 19 features in women) from multiple aspects. The input features were selected based on their easy availability and known relevance to bone health. Most of the selected features showed significance comparing the decreased bone density group and the normal bone density group in our study population. 

The increase in the number of input features may raise concerns about the applicability of the models in clinical practice. From this aspect, the traditional model OSTA seems to be the most convenient tool, since it only requires simple calculations with age and weight. However, with the aids of automated calculator and alert system, this problem can be solved, and can potentially call more attention from the patients and physicians to recognize the risk of osteoporosis. As mentioned previously, osteoporosis is treatable, but often overlooked. By incorporating the prediction models in clinical practice, patients could benefit from earlier diagnosis and intervention. 

To assess the contribution of the input features, we made smaller models with six variables, including age, weight, history of smoking, history of alcohol drinking, ALT, creatinine, using the same methods proposed, to compare with the full models. These variables were selected for their availability and larger effect size observed in our study population. The results were summarized in the Appendix A. As the results suggested, reducing the number of variables resulted in poorer performance, although statistical significances were difficult to establish. By comparison of the full models and the smaller 6-variable models, it is evident that inclusion of these variables provided more information, and contributed to the better performance in the machine learning models.

For the determination of cutoff points in different models, we adopted weighted Youden index at weight 0.6 [31]. In ideal conditions, the cutoff points of ROC curves could be decided by cost and benefit analysis [35]; however, the cost and benefit of different conditions were difficult to quantify. As an alternative, weighted Youden index was designed to balance sensitivity and specificity for different requirements [31,36]. In our study, we aimed to construct screening tools. The cost of false positive would mainly be that the patient underwent DXA to confirm the presence of osteoporosis; however, the cost of false negative was that osteoporosis would be unrecognized, left untreated, and osteoporotic fracture could not be prevented. Therefore, instead of putting the same weight 0.5 on both sensitivity and specificity (which was, the original Youden index), we tested and set the weight to 0.6 to emphasize more on the sensitivity, and generate more suitable cutoff value for real-world application. The cutoff value for the OSTA model in our study was also calculated and resulted in <3 in men and <0 in women. The performance of the OSTA model, and the corresponding sensitivity and specificity were compatible to previous studies [37]. In previous study, the OSTA model reported an AUROC of 0.79, with 91% sensitivity and 45% specificity [20].

Several limitations of the study should be considered. First, some of the input features in our study were recorded by history taking and were prone to recall bias. Additionally, the results of DXA in the database were categorical, being reported as normal bone density, osteopenia, and osteoporosis. If we could gain the information of the original *T* score of the tests, continuous modelling could be made possible, and the performance could be further improved. Second, selection bias might occur during inclusion of participants. When comparing with other studies [1,5,21,26], the prevalence of osteoporosis was lower in our study population, being 3.8% in men and 10.4% in women. This difference in prevalence might be related to the lower average age and better health condition in our participants. To overcome the problem of imbalanced class proportion of categorical outcomes, we trained the models with categorical groupings different from that in the testing process. This should not cast a problem since all the subsequent statistical analysis were done with osteoporosis being the prediction target. Finally, characteristics and risk factors in osteoporosis differ among ethnicity groups and different environments. The validity of the machine learning models should be examined for different populations worldwide.

## 5. Conclusions

Prediction models using machine learning algorithms including ANN, SVM, RF, KNN, and LoR were constructed with easily available input features to serve as screening tools for osteoporosis in community-dwelling men and women older than 50 years old. The models reached AUROC of 0.821–0.843 in men and 0.767–0.811 in women. At designated cutoff points, the models also achieved 83–96% sensitivity and 53–73% specificity in men; 76–90% sensitivity and 62–69% specificity in women. Our results for the machine learning model generally outperformed the traditional model, OSTA. We have shown that machine learning could improve the screening performance in osteoporosis. It would be difficult to choose a definite winner between the models due to their similar performances and applications in different populations and situations. By incorporating the models in clinical practice, both the patients and physicians could be more aware of the disease, and potentially benefit from an earlier diagnosis and treatment of osteoporosis.

## Figures and Tables

**Figure 1 ijerph-18-07635-f001:**
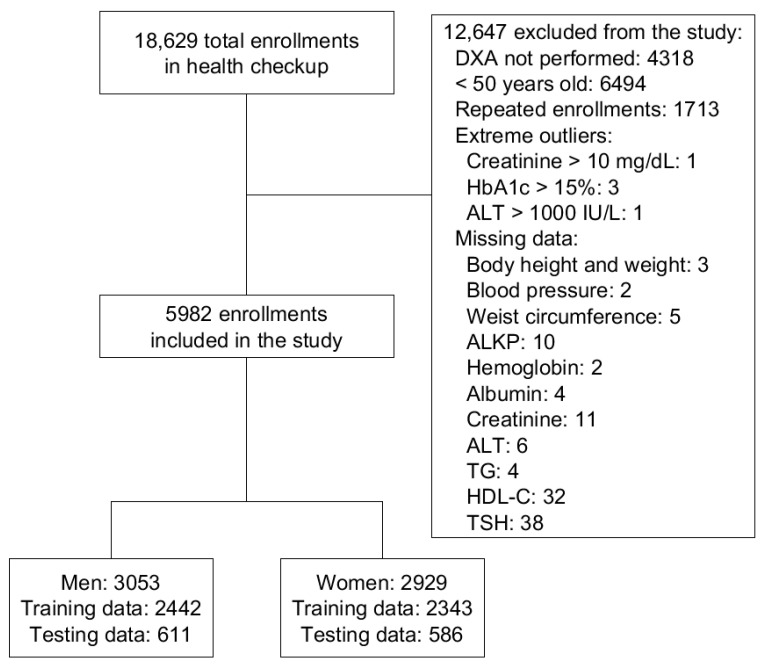
Flowchart of data inclusion and preprocessing. DXA: dual-energy *X*-ray absorptiometry; HbA1c: Hemoglobin A1c; ALT: alanine transaminase; ALK-P: alkaline phosphatase; TG: triglyceride; HDL-C: high-density lipoprotein cholesterol; TSH: thyroid-stimulating hormone.

**Figure 2 ijerph-18-07635-f002:**
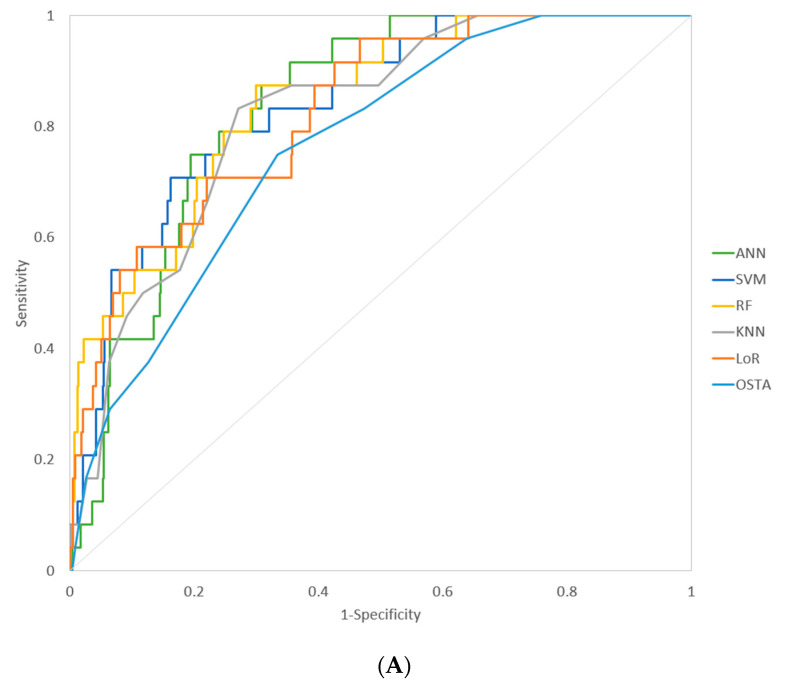
(**A**)The ROC curves of different machine learning models and the OSTA model for prediction of osteoporosis in men. ANN: Artificial neural network; SVM: Support vector machine; RF: Random Forest; KNN: K-nearest neighbors; LoR: Logistic regression; OSTA: Osteoporosis Self-assessment Tool for Asian; ROC curve: Receiver operating characteristic curve. (**B**) The ROC curves of different machine learning models and the OSTA model for prediction of osteoporosis in women.

**Table 1 ijerph-18-07635-t001:** Selection of hyperparameters for the ANN, SVM, RF, and KNN models.

Hyperparameter	Examined Range *	Selected Hyperparameter for the Men Model	Selected Hyperparameter for the Women Model
ANN			
Number of hidden layers	1–2	2	2
Number of nodes	4–20 in each hidden layer	9 in hidden layer 1, 4 in hidden layer 2	13 in hidden layer 1, 7 in hidden layer 2
Learning rate	0.01–0.0001	0.001	0.001
Dropout rate	0–60%	40%	40%
SVM			
Kernel type	linear, polynomial, or radial basis function	radial basis function	radial basis function
Regularization parameter C	2^−2^–2^9^	2^5^	2^4^
Kernel coefficient gamma	2^−9^–2^2^	2^−3^	2^−1^
Degree for Polynomial Function	1–4	-	-
RF			
Number of trees	100–1000	300	600
Number of features to consider	3–11	8	10
Maximum depth of the tree	3–11	8	10
KNN			
Number of neighbors	1–30	28	13
Leaf size	1–49	15	3
Power parameter *p*	Manhattan distance or Euclidean distance	Manhattan distance	Euclidean distance

* Within the examined range, different combinations of hyperparameters were tested for the ANN model; grid search was done for the SVM, RF, and KNN models. ANN: Artificial neural network; SVM: Support vector machine; RF: Random Forest; KNN: K-nearest neighbors.

**Table 2 ijerph-18-07635-t002:** Baseline characteristics of the study population.

Characteristic	Total (n = 5982)	Men (n = 3053)	Women (n = 2929)	*p*-Value *
Age (years)	59.3 ± 7.0	59.3 ± 7.0	59.3 ± 7.0	0.9650
Body height (cm)	161.8 ± 8.4	167.7 ± 6.2	155.7 ± 5.5	<0.0001
Body weight (kg)	63.5 ± 11.7	69.8 ± 10.2	56.9 ± 9.2	<0.0001
Waist circumference (cm)	83.9 ± 9.8	87.8 ± 8.4	79.9 ± 9.5	<0.0001
History of smoking (n, %)	1255 (21.0)	1133 (37.1)	122 (4.2)	<0.0001
History of alcohol drinking (n, %)	484 (8.1)	414 (13.6)	70 (2.4)	<0.0001
Diabetes mellitus (n, %)	1075 (18.0)	618 (20.2)	457 (15.6)	<0.0001
Hypertension (n, %)	2226 (37.2)	1250 (40.9)	976 (33.3)	<0.0001
Albumin (g/dL)	4.50 ± 0.30	4.52 ± 0.27	4.45 ± 0.26	<0.0001
Hemoglobin (g/dL)	14.1 ± 1.4	15.0 ± 1.2	13.2 ± 1.1	<0.0001
ALT (IU/L)	27.5 ± 18.6	30.5 ± 20.0	24.4 ± 16.6	<0.0001
Creatinine (mg/dL)	0.89 ± 0.27	1.03 ± 0.26	0.75 ± 0.21	<0.0001
TG (mg/dL)	133.1 ± 84.8	147.2 ± 95.2	118.4 ± 69.1	<0.0001
HDL-C (mg/dL)	55.1 ± 16.4	48.76 ± 13.54	61.78 ± 1.57	<0.0001
ALK-P (IU/L)	67.9 ± 19.9	66.0 ± 18.4	69.9 ± 21.2	<0.0001
TSH (uIU/mL)	2.32 ± 2.34	2.18 ± 2.15	2.47 ± 2.52	<0.0001
Menopause (n, %)			2448 (83.6)	
History of HRT (n, %)			283 (9.7)	
Parity (n)			2.4 ± 1.4	
Categories of bone density result				<0.0001
Normal (n, %)	3058 (51.1)	1802 (59.0)	1256 (42.9)	
Osteopenia (n, %)	2503 (41.8)	1134 (37.1)	1369 (46.7)	
Osteoprosis (n, %)	421 (7.0)	117 (3.8)	304 (10.4)	

* *p*-values were calculated with two-tailed *T* tests for continuous variables, two-tailed *Z* tests for binary variables, and Chi-square tests for categorical variables. ALT: alanine transaminase; TG: triglyceride; HDL-C: high-density lipoprotein cholesterol; ALK-P: alkaline phosphatase; TSH: thyroid-stimulating hormone; HRT: hormone-replacement therapy.

**Table 3 ijerph-18-07635-t003:** Comparison of the input features between participants with normal bone density and decreased bone density.

Feature	Normal Bone Density (n = 3058, 51.1%)	Decreased Bone Density * (n = 2924, 48.9%)	*p*-Value **
Age (years)	57.6 ± 6.1	61.1 ± 7.3	<0.0001
Body height (cm)	163.9 ± 8.0	159.6 ± 8.1	<0.0001
Body weight (kg)	67.1 ± 11.5	59.8 ± 10.6	<0.0001
Waist circumference (cm)	85.7 ± 9.5	82.1 ± 9.6	<0.0001
History of smoking (n, %)	739 (24.2)	516 (17.7)	<0.0001
History of alcohol drinking (n, %)	291 (9.5)	193 (6.6)	<0.0001
Diabetes mellitus (n, %)	550 (18.0)	525 (18.0)	0.9753
Hypertension (n, %)	1151 (37.6)	1075 (36.8)	0.4844
Albumin (g/dL)	4.49 ± 0.26	4.47 ± 0.28	0.0289
Hemoglobin (g/dL)	14.3 ± 1.5	13.9 ± 1.4	<0.0001
ALT (IU/L)	28.9 ± 19.4	26.0 ± 17.7	<0.0001
Creatinine (mg/dL)	0.92 ± 0.26	0.86 ± 0.28	<0.0001
TG (mg/dL)	138.2 ± 84.3	127.8 ± 85.0	<0.0001
HDL-C (mg/dL)	52.9 ± 15.6	57.5 ± 16.9	<0.0001
ALK-P (IU/L)	64.9 ± 17.9	71.1 ± 21.5	<0.0001
TSH (uIU/mL)	2.32 ± 2.05	2.32 ± 2.63	0.9480
Abnormal TSH level (<0.01 or ≥4.5 uIU/mL) (n, %)	222 (7.3)	253 (8.7)	0.0464
Menopause (n, %)	937 (74.6)	1511 (90.3)	<0.0001
History of HRT (n, %)	127 (10.1)	156 (9.3)	0.4756
Parity	2.24 ± 1.25	2.60 ± 1.55	<0.0001

* Decreased bone density referred to participants with osteopenia or osteoporosis. ** *p*-values were calculated with two-tailed *T* tests for continuous variables, and two-tailed *Z* tests for binary variables. ALT: alanine transaminase; TG: triglyceride; HDL-C: high-density lipoprotein cholesterol; ALK-P: alkaline phosphatase; TSH: thyroid-stimulating hormone; HRT: hormone-replacement therapy.

**Table 4 ijerph-18-07635-t004:** Performance of different machine learning models and the OSTA model for prediction of osteoporosis in men and women.

Model	AUROC (95% CI)	Sensitivity *	Specificity *	*p*-Value ** (Compare with OSTA)
Men				
ANN	0.837 (0.805–0.865)	0.917	0.646	0.0151
SVM	0.840 (0.809–0.868)	0.917	0.547	0.0061
RF	0.843 (0.812–0.871)	0.875	0.700	0.0321
KNN	0.821 (0.788–0.851)	0.833	0.729	0.1087
LoR	0.827 (0.794–0.856)	0.958	0.533	0.0421
OSTA ***	0.766 (0.730–0.799)	0.958	0.361	
Women				
ANN	0.781 (0.745–0.814)	0.864	0.643	0.0258
SVM	0.807 (0.773–0.838)	0.898	0.651	0.0136
RF	0.811 (0.777–0.842)	0.898	0.624	0.0006
KNN	0.767 (0.731–0.801)	0.762	0.692	0.3563
LoR	0.772 (0.732–0.806)	0.814	0.670	0.0808
OSTA***	0.734 (0.697–0.770)	0.763	0.630	

* Sensitivity and specificity were based on cutoff values calculated by the weighted Youden index with weight set at 0.6. ** *p*-values were calculated with the nonparametric method to compare two ROC curves proposed by DeLong et al. *** Sensitivity and specificity for the OSTA model were obtained at cutoff value <3 for men models, <0 for women models. ANN: Artificial neural network; SVM: Support vector machine; RF: Random Forest; KNN: K-nearest neighbors; LoR: Logistic regression; OSTA: Osteoporosis Self-assessment Tool for Asian; AUROC: Area under the receiver operating characteristic curve; CI: Confidence interval; ROC curve: receiver operating characteristic curve.

## Data Availability

All data generated or analyzed during this study can be included in this published article. Due to legal restrictions imposed by the government of Taiwan in relation to the “Personal Information Protection Act”, data cannot be made publicly available. The data that support the findings of this study are available from the corresponding author upon reasonable request.

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
