# Peer review of "Development of Machine Learning Models for Prediction of Osteoporosis from Clinical Health Examination Data"

_ijerph, 2021, doi:10.3390/ijerph18147635_

Round 1
Reviewer 1 Report
There are a number of important points to consider:
- Lines 112-122: From the present discussion, it is not clear why the continuous variable 'bone density' is not used from the beginning. Please justify this point.
- By way of contrast, it would be interesting to consider a joint model that includes the sex variable.
- Lines 154-167: It is necessary to justify the selection of the parameters of the different models.
- Lines 259- 261: Models built with ML in a similar way to the one developed in the paper should be compared with the variables of the OSTA model to compare the results and assess the real contribution of the model finally proposed by the authors.
- One of the main problems is that the data are inaccessible to other researchers, not even anonymized, so how can the results that the authors claim to have obtained be checked, studied or improved? Reproducibility is an essential part of the scientific process, so the work should not be published as long as the data cannot be used by other researchers (once the sensitive information has been removed) or the authors should use the proposed methodology on data that are accessible, even if only on reasoned request.
- In order to test the validity and scope of the proposed model, it is necessary to carry out comparative studies with similar models constructed with a smaller number of independent variables.
Other minor observations should be taken into account (especially to ensure that all relevant information is contained in the article without the need to resort to references unless it is desired to expand the information substantially).
- Line 78: Briefly explain the T-score index used.
- Section 2.2: In the description of the variables (lines 101-111) the relationship with the dependent variable must be explained (perhaps the simplest way would be to use the results of Table 3, developing them).
- Lines 137-141: The way to limit the overfitting from the truncation in the number of interactions must be justified theoretically. Specifically the way to select the stopping point.
- Line 183: Develop the concept of Youden index, justifying its use and the selection of 0.6.
- Tables 1 and 2: Explain the p-value used. Even if it can be deduced to which contrast it belongs, it should be explicitly stated.
Author Response
Response to Reviewer 1 Comments
Thank you for your letter and constructive comments concerning our manuscript entitled “Development of Machine Learning Models for Prediction of Osteoporosis from Clinical Health Examination Data”. We have studied your comments carefully and made corresponding revision which we hope to meet your suggestions. The response to your questions or comments were shown in details in the following texts:
Point 1: Lines 112-122: From the present discussion, it is not clear why the continuous variable 'bone density' is not used from the beginning. Please justify this point.
Response 1: Thank you for your suggestion. 5982 participants in our study performed BMD examination with DXA. The BMD data were interpreted by qualified radiology specialists to three categorical groups as osteoporosis (T-score ≤ -2.5), osteopenia (-2.5 < T-score < -1), or normal (T-score ≥ -1) in the final report card. We agree that, with the original T score or bone density, continuous modelling could be made possible, and the performance could be further improved. However, the information of the continuous bone density value was not available in our database. We have recognized this as our limitation on Lines 320-324. (Besides, the results of DXA in the database were categorical, being reported as normal bone density, osteopenia, and osteoporosis. If we could gain the information of the original T score of the tests, continuous modelling could be made possible, and the performance could be further improved.)
Point 2: By way of contrast, it would be interesting to consider a joint model that includes the sex variable.
Response 2: Thank you for the suggestion for comparing our models to the joint model including sex as a variable. We have reviewed previous studies focusing on osteoporosis pathophysiology between male and female genders. Among male patients, only 30 to 50% cases have primary osteoporosis; secondary causes including alcohol abuse, steroid treatment and other metabolic disorders. Besides, prevalence of secondary osteoporosis in women was much lower as compared to men. Postmenopausal estrogen deficiency and senile osteoporosis stands for the main causes of primary osteoporosis in women (1). Due to the difference in pathophysiology, risk factors, and baseline characteristics, we trained the prediction models for men and women separately to fit the model better to the target population. We have clarified the statement on Lines 274-280. (Among male patients, only 30 to 50% cases have primary osteoporosis; secondary causes including alcohol abuse, steroid treatment and other metabolic disorders. Besides, prevalence of secondary osteoporosis in women was much lower as compared to men. Postmenopausal estrogen deficiency and senile osteoporosis stands for the main causes of primary osteoporosis in women. Due to the difference in etiology and baseline characteristics, we trained prediction models for men and women separately, and gained satisfactory results.)
Point 3: Lines 154-167: It is necessary to justify the selection of the parameters of the different models.
Response 3: Thank you for the valuable suggestion. For hyperparameter tuning, we have tested different combination of the number of layers, number of nodes, learning rate, and dropout rate in the ANN model. In the SVM, RF, KNN models, grid search for the hyperparameters was performed to find out the combination that yield the best performance. To be more specific, in the ANN model, we tested with 1 or 2 hidden layers, adjusted the number of nodes from 4 to 20, testing the learning rate from 0.01 to 0.0001, and dropout rate from 0% to 60%. In the SVM model, kernel type with linear, polynomial, and RBF, regularization parameter C from 2-2 to 29, kernel coefficient gamma from 2-9 to 22, and degree for polynomial kernel function from 1 to 4 were grid-searched. For the RF model, the number of tree from 100 to 1000, number of features to consider from 3 to 11, and the maximum depth of the tree from 3 to 11 were grid-searched. For the KNN models, the number of neighbors from 1 to 30, the leaf size from 1 to 49, and the power parameter p with Manhattan distance and Euclidean distance were grid-searched. The set of hyperparameters that gain the best performance from the validation data was selected for the models. We have clarified the process on Lines 154-169, and presented the final result in Table 1. (For hyperparameter tuning, different combination of the number of layers (1 or 2 hidden layers), number of nodes (from 4 to 20 in each hidden layer), learning rate (from 0.01 to 0.0001), and dropout rate (from 0% to 60%) were tested for the ANN model. Grid search was done for the SVM, RF, and KNN model. For the SVM model, the kernel type (linear, polynomial, or radial basis function), regularization parameter C (from 2-2 to 29), kernel coefficient gamma (from 2-9 to 22), and degree (from 1 to 4) for the polynomial kernel function were examined; for the RF model, the number of trees (from 100 to 1000), number of features to consider (from 3 to 11), and the maximum depth of the tree (from 3 to 11) were examined; for the KNN model, the number of neighbors (from 1 to 30), the leaf size (from 1 to 49), and the power parameter p (Manhattan distance or Euclidean distance) were tested. All models were constructed with a balanced class weight. For each set of hyperparameter, randomization for the split of validation dataset and the training process were repeated several times. Area under the receiver operating characteristic curve (AUROC) was calculated for the validation dataset in each training process, and the mean value of AUROC were com-pared. We adopted the hyperparameter that yield a better mean AUROC during validation to train our models.)
|
Table 1-1. Selection of hyperparameters for the ANN model. |
|
|
||
|
|
Number of hidden layers |
Number of nodes |
Learning rate |
Dropout rate |
|
Examine range |
1-2 |
4-20 in each hidden layer |
0.01-0.0001 |
0-60% |
|
Selected hyperparameters for the men model |
2 |
9 in hidden layer 1, 4 in hidden layer 2 |
0.001 |
40% |
|
Selected hyperparameters for the women model |
2 |
13 in hidden layer 1, 7 in hidden layer 2 |
0.001 |
40% |
|
ANN: Artificial neural network |
||||
|
Table 1-2. Selection of hyperparameters for the SVM model. |
|
|
||
|
|
Kernal type |
Regularization parameter C |
Kernel coefficient gamma |
Degree for polynomial function |
|
Grid search range |
linear, polynomial, or radial basis function |
2-2-29 |
2-9-22 |
1-4 |
|
Selected hyperparameters for the men model |
radial basis function |
25 |
2-3 |
- |
|
Selected hyperparameters for the women model |
radial basis function |
24 |
2-1 |
- |
|
SVM: Support vector machine |
||||
|
Table 1-3. Selection of hyperparameters for the RF model. |
|
||
|
|
Number of trees |
Number of features to consider |
Maximum depth of the tree |
|
Grid search range |
100-1000 |
3-11 |
3-11 |
|
Selected hyperparameters for the men model |
300 |
8 |
8 |
|
Selected hyperparameters for the women model |
600 |
10 |
10 |
|
RF: Random forest |
|||
|
Table 1-4. Selection of hyperparameters for the KNN model. |
|
||
|
|
Number of neighbors |
Leaf size |
Power parameter p |
|
Grid search range |
1-30 |
1-49 |
Manhattan distance or Euclidean distance |
|
Selected hyperparameters for the men model |
28 |
15 |
Manhattan distance |
|
Selected hyperparameters for the women model |
13 |
3 |
Euclidean distance |
|
KNN: K-nearest neighbors |
|||
Point 4: Lines 259- 261: Models built with ML in a similar way to the one developed in the paper should be compared with the variables of the OSTA model to compare the results and assess the real contribution of the model finally proposed by the authors.
Response 4: Thank you for the suggestion. Variables used in the OSTA model were age and weight. To validate the contribution of other variables in our study, we developed similar models using the same methods with a smaller set of variables: age, weight, history of smoking, history of alcohol drinking, ALT, and creatinine (this included age and weight as in the OSTA model). These variables were selected due to their easy availability and larger effect size in our study population. The results were summarized in the Supplementary Table S2.
|
Table S2. Performance of the full models and the smaller models with 6 variables for prediction of osteoporosis in men and women. |
|||
|
|
AUROC of full models (95% CI) |
AUROC of models with 6 variables (95% CI) |
p-value* |
|
Men |
|
|
|
|
ANN |
0.837 (0.805-0.865) |
0.779 (0.744-0.812) |
0.0682 |
|
SVM |
0.840 (0.809-0.868) |
0.812 (0.779-0.842) |
0.2526 |
|
RF |
0.843 (0.812-0.871) |
0.812 (0.779-0.843) |
0.2725 |
|
KNN |
0.821 (0.788-0.851) |
0.775 (0.740-0.807) |
0.1388 |
|
LoR |
0.827 (0.794-0.856) |
0.782 (0.747-0.814) |
0.0153 |
|
|
|
|
|
|
Women |
|
|
|
|
ANN |
0.781 (0.745-0.814) |
0.762 (0.726-0.796) |
0.1704 |
|
SVM |
0.807 (0.773-0.838) |
0.783 (0.747-0.816) |
0.3643 |
|
RF |
0.811 (0.777-0.842) |
0.785 (0.749-0.817) |
0.1778 |
|
KNN |
0.767 (0.731-0.801) |
0.774 (0.738-0.807) |
0.8242 |
|
LoR |
0.772 (0.732-0.806) |
0.761 (0.724-0.795) |
0.4470 |
|
*P-values were calculated with the nonparametric methods to compare two ROC curves proposed by DeLong et al. ANN: Artificial neural network; SVM: Support vector machine; RF: Random forest; KNN: K-nearest neighbors; LoR: Logistic regression; AUROC: Area under the receiver operating characteristic curve; CI: Confidence interval; ROC curve: receiver operating characteristic curve. |
|||
In most of the models, reducing the number of variables resulted in poorer performance, although statistical significances were difficult to establish. The variables we included in the study were known factors associated with osteoporosis, and most showed significant difference in the normal BMD group and decreased BMD group, as Table 3 suggested. By comparison of the full models and the smaller 6-variable models, it is evident that inclusion of these variables provided more information, and contributed to the better performance in the machine learning models. We have revised our manuscript to include the content on Lines 294-302. (To assess the contribution of the input features, we made smaller models with six variables, including age, weight, history of smoking, history of alcohol drinking, ALT, creatinine, using the same methods proposed, to compare with the full models. These variables were selected for their availability and larger effect size observed in our study population. The results were summarized in the Supplementary Table S2. As the results suggested, reducing the number of variables resulted in poorer performance, although statistical significances were difficult to establish. By comparison of the full models and the smaller 6-variable models, it is evident that inclusion of these variables provided more information, and contributed to the better performance in the machine learning models.)
Point 5: One of the main problems is that the data are inaccessible to other researchers, not even anonymized, so how can the results that the authors claim to have obtained be checked, studied or improved? Reproducibility is an essential part of the scientific process, so the work should not be published as long as the data cannot be used by other researchers (once the sensitive information has been removed) or the authors should use the proposed methodology on data that are accessible, even if only on reasoned request.
Response 5: Thank you for the suggestion. We have revised our data sharing policy in the “Data Availability Statement” on Lines 365-369. (All data generated or analyzed during this study can be included in this published article. Due to legal restrictions imposed by the government of Taiwan in relation to the “Personal Information Protection Act”, data cannot be made publicly available. The data that support the findings of this study are available from the corresponding author upon reasonable request.)
Point 6: In order to test the validity and scope of the proposed model, it is necessary to carry out comparative studies with similar models constructed with a smaller number of independent variables.
Response 6: Thank you for the suggestion. As in “Response 4”, we have conducted an experiment to include age, weight, history of smoking, history of alcohol drinking, ALT, and creatinine as variables, and constructed machine learning models in similar methods. The results and discussion were presented in “Response 4”.
Minor
Point 1: Line 78: Briefly explain the T-score index used.
Response 1: Thank you for the delicate suggestion. We have revised and explained T-score index more clearly on Lines 77-83. (BMD was checked with DXA (Lunar Prodigy Advance; GE Healthcare, Madison, WI, USA) at the lumbar spine and bilateral hip joint. T-score represented the standard deviation of BMD from the healthy young adults of same sex and ethnicity. The BMD results were compared with the results from healthy young adults. The T-scores at each site were obtained, and the lowest T-score was used to interpret the results as osteoporosis (T-score ≤ -2.5), osteopenia (-2.5 < T-score < -1), or normal (T-score ≥ -1) in the final report card.)
Point 2: Section 2.2: In the description of the variables (lines 101-111) the relationship with the dependent variable must be explained (perhaps the simplest way would be to use the results of Table 3, developing them).
Response 2: Thank you for the comment. In our study, we have reviewed risk factors for osteoporosis in the Introduction section, including age, female gender, low body weight, low physical activity, poor nutritional status, and several endocrine and cardiometabolic factors (2-10). Aside from applying well-studied factors as input features to our machine learning models, the relationship between input features and osteoporosis were examined in our study population in Table 3, and most showed significant difference in the normal BMD group and the decreased BMD group. We have included the content on Lines 283-285. (The input features were selected based on their easy availability and known relevance to bone health. Most of the selected features showed significance comparing the decreased bone density group and the normal bone density group in our study population.)
Point 3: Lines 137-141: The way to limit the overfitting from the truncation in the number of interactions must be justified theoretically. Specifically the way to select the stopping point.
Response 3:. Thank you for the suggestion. In machine learning models, overfitting occurs when the models learn from the noise in the training data instead of the general relation between inputs and outputs. This would interfere with the training process and the final performance of the constructed models. Several methods were proposed to reduce the impact of overfitting (11). One of the methods is network reduction. By limiting the size of model, learning from less meaningful, irrelevant interactions could be avoided. Early stopping is an important way in ANN to reduce overfitting. To be specific, we monitored the performance (the AUROC) of the training and validation dataset in each epoch. Training was stopped when the AUROC of the validation dataset reached a peak and did not show further improvement after 100 epochs. The model generated from the peak point would be used for further analysis. We have clarified the process on Lines 148-153. (To avoid overfitting, we limited the complexity of the models, and applied early stopping and dropout during training of the ANN model. For early stopping, we monitored the AUROC of the training and validation dataset in each epoch. Training was stopped when the AUROC of the validation dataset reached a peak and did not show further improvement after 100 epochs. The model generated from the peak point was used for further analysis.)
Point 4: Line 183: Develop the concept of Youden index, justifying its use and the selection of 0.6.
Response 4: Thank you for the comment. There are many ways to determine the optimal cut-off point in ROC curves. The more popular ways included the point closest to (0,1), Youden index, and minimize cost (12). Youden index maximizes the vertical distance from the point to the line x=y, which also maximizes the sum of sensitivity and specificity. It is widely used for its simplicity, and provides a quick understanding about the performance of the diagnostic tool. In contrast, to optimize a diagnostic tool in real world situation, the prevalence of disease, the role of the tool, and the cost and effect of different results (the true positive, true negative, false positive, and false negative) should be considered (13). The most precise method is to calculate and quantify the cost of different results, then search for the point that minimize the cost. However, this would be strenuous and difficult to accomplish. As an alternative, putting weight on Youden index is a way to emphasize more on one end of sensitivity or specificity (13-15). In our study, we aimed to construct screening tools. The cost of false positive would mainly be that the patient undergo DXA to confirmed the presence of osteoporosis; however, the cost of false negative is that osteoporosis would be unrecognized, left untreated, and osteoporotic fracture could not be prevented. Therefore, putting the same weight 0.5 on both sensitivity and specificity (which is, the original Youden index) would not help to let our readers understand the potential performance of the models in real world situation. The desired weight for different ROC curves is still under investigation, and would probably need to be considered case-by-case. We referred to the study performed by Li et al (16) and selected 0.6, which was one of the value tested, to put the concept into practice. We have discussed the concept and revised the manuscript on Lines 303-314. (For the determination of cutoff points in different models, we adopted weighted Youden index at weight 0.6. In ideal conditions, the cutoff points of ROC curves could be decided by cost and benefit analysis; however, the cost and benefit of different conditions were difficult to quantify. As an alternative, weighted Youden index was designed to balance sensitivity and specificity for different requirements. In our study, we aimed to construct screening tools. The cost of false positive would mainly be that the patient underwent DXA to confirmed the presence of osteoporosis; however, the cost of false negative was that osteoporosis would be unrecognized, left untreated, and osteoporotic fracture could not be prevented. Therefore, instead of put-ting the same weight 0.5 on both sensitivity and specificity (which was, the original Youden index), we tested and set the weight to 0.6 to emphasize more on the sensitivity, and generate more suitable cutoff value for real world application.)
Point 5: Tables 1 and 2: Explain the p-value used. Even if it can be deduced to which contrast it belongs, it should be explicitly stated.
Response 5: Thank you for the suggestion. We have revised our manuscript and explained the tests applied in Table 2, Table 3, and Table 4 in the corresponding footnotes.
- Pietschmann P, Kerschan-Schindl K. Osteoporosis: gender-specific aspects. Wien Med Wochenschr. 2004;154(17-18):411-5.
- Bijelic R, Milicevic S, Balaban J. Risk Factors for Osteoporosis in Postmenopausal Women. Med Arch. 2017;71(1):25-8.
- Kelsey JL. Risk factors for osteoporosis and associated fractures. Public Health Rep. 1989;104 Suppl(Suppl):14-20.
- Delitala AP, Scuteri A, Doria C. Thyroid Hormone Diseases and Osteoporosis. J Clin Med. 2020;9(4).
- Compston JE, McClung MR, Leslie WD. Osteoporosis. Lancet. 2019;393(10169):364-76.
- NIH Consensus Development Panel on Osteoporosis Prevention, Diagnosis, and Therapy, March 7-29, 2000: highlights of the conference. South Med J. 2001;94(6):569-73.
- Ilić K, Obradović N, Vujasinović-Stupar N. The relationship among hypertension, antihypertensive medications, and osteoporosis: a narrative review. Calcif Tissue Int. 2013;92(3):217-27.
- Kanazawa I. Interaction between bone and glucose metabolism [Review]. Endocr J. 2017;64(11):1043-53.
- Pan ML, Chen LR, Tsao HM, Chen KH. Iron Deficiency Anemia as a Risk Factor for Osteoporosis in Taiwan: A Nationwide Population-Based Study. Nutrients. 2017;9(6).
- Aspray TJ, Hill TR. Osteoporosis and the Ageing Skeleton. Subcell Biochem. 2019;91:453-76.
- Ying X, editor An overview of overfitting and its solutions. Journal of Physics: Conference Series; 2019: IOP Publishing.
- Kumar R, Indrayan A. Receiver operating characteristic (ROC) curve for medical researchers. Indian pediatrics. 2011;48(4):277-87.
- Baker SG, Kramer BS. Peirce, Youden, and receiver operating characteristic curves. The American Statistician. 2007;61(4):343-6.
- Schisterman EF, Faraggi D, Reiser B, Hu J. Youden Index and the optimal threshold for markers with mass at zero. Statistics in medicine. 2008;27(2):297-315.
- Rücker G, Schumacher M. Summary ROC curve based on a weighted Youden index for selecting an optimal cutpoint in meta‐analysis of diagnostic accuracy. Statistics in medicine. 2010;29(30):3069-78.
- Li DL, Shen F, Yin Y, Peng JX, Chen PY. Weighted Youden index and its two-independent-sample comparison based on weighted sensitivity and specificity. Chin Med J (Engl). 2013;126(6):1150-4.

Reviewer 2 Report
The paper "Development of Machine Learning Models for Prediction of Osteoporosis from Clinical Health Examination Data" present the prediction models with machine learning algorithms to serve as screening tools for osteoporosis in adults over fifty years old. The comparation the performance of newly-developed models with traditional prediction models is achieved. Data was acquired from community-dwelling participants en- rolled in health checkup programs at a medical center in Taiwan.
Some sugestions:
- The references can be updated with the research in the last 5 years.
- Line 205 must be splited in 2 sentences.
- Conclusion can be improved
Author Response
Response to Reviewer 2 Comments
Thank you for your letter and constructive comments concerning our manuscript entitled “Development of Machine Learning Models for Prediction of Osteoporosis from Clinical Health Examination Data”. We have studied your comments carefully and made corresponding revision which we hope to meet your suggestions. The response to your questions or comments were shown in details in the following texts:
Point 1: The references can be updated with the research in the last 5 years.
Response 1: Thank you for the suggestion. We have updated some latest references about risk factors and application of machine learning models in osteoporosis. The reference 6, 9, 19, 34 were added, and content in manuscript was revised accordingly. (Ref 6: Johnston CB, Dagar M. Osteoporosis in Older Adults. Med Clin North Am. 2020;104(5):873-84.; Ref 9: Choksi P, Jepsen KJ, Clines GA. The challenges of diagnosing osteoporosis and the limitations of currently available tools. Clinical Diabetes and Endocrinology. 2018;4(1):12.; Ref 19: Aspray TJ, Hill TR. Osteoporosis and the Ageing Skeleton. Subcell Biochem. 2019;91:453-76.; Ref 34: Pietschmann P, Kerschan-Schindl K. Osteoporosis: gender-specific aspects. Wien Med Wochenschr. 2004;154(17-18):411-5.)
Point 2: Line 205 must be split in 2 sentences.
Response 2: Thank you for the suggestion. We have split the sentence on Lines 229-232. (In the five machine learning models (ANN, SVM, RF, KNN, LoR) and the traditional model (OSTA), the AUROC, sensitivity, and specificity were presented in Table 4. The sensitivity and specificity were calculated at the cutoff value determined by maximization of weighted Youden index at weighted 0.6.)
Point 3: Conclusion can be improved
Response 3: Thank you for the comment. We have modified our conclusion according to your advice and added new content on Lines 255-262 (Prediction models using machine learning algorithms including ANN, SVM, RF, KNN, and LoR were constructed with easily available input features to serve as screening tools for osteoporosis in community-dwelling men and women older than 50 years old. The models reached AUROC of 0.821-0.843 in men and 0.767-0.811 in women. At designated cutoff points, the models also achieved 83-96% sensitivity and 53-73% specificity in men; 76-90% sensitivity and 62-69% specificity in women. Our results of machine learning model generally out-performed the traditional model OSTA. We have shown that machine learning could improve the screening performance in osteoporosis. It would be difficult to choose a definite winner of the models due to similarity in performance and application in different population and different situations. By incorporating the models in clinical practice, both the patients and physicians could be more aware of the disease, and potentially benefit from earlier diagnosis and treatment of osteoporosis.)

Reviewer 3 Report
Reviewer's comments on paper “Development of Machine Learning Models for Prediction of Osteoporosis from Clinical Health Examination Data” submitted to Int. J. Environ. Res. Public Health (ijerph-1258605).
Although some interesting results can be observed, some aspects should be improved before publication. Please see specific comments listed below for details. There is a few of weaknesses in the articles which are mentioned below in the detailed comments.
L35 Please cite source reference, as [5] is about US population.
L39 Since 2017 proper abbreviation dual-energy x-ray absorptiometry is DXA not DEXA. Please correct throughout the manuscript.
L42 Correct “bone marrow density” to “bone mineral density”
L92 Suggestion – Is it possible to categorize these inputs into some groups ?
L99 Please provide some details about this "careful examination”. What was the procedure? Who performed selection? Single person or some council? Physician? If yes, of what specialization ?
L127 What do you mean be “rescaled” ? Normalized ?
L154 I suggest adding the table in which all machine models are characterized - this will allow for a clearer characterization of the models used and to will help to visualize the differences between models used for men and women datasets.
L174 ROC abbreviation was nod defined earlier (only AUROC, L151)
Tables: the information which tests were used to calculate p-values should be given in each table footnote.
Table 1: give the p-values for normal and osteopenic bone densities
L232-236 Repetition of the results - please remove
L237-239 Is sound as final conclusion rather than one of the first sentences of the discussion.
L293 Which machine learning model seems to be the most promising tool? Is there a winner ?
Author Response
Response to Reviewer 3 Comments
Thank you for your letter and constructive comments concerning our manuscript entitled “Development of Machine Learning Models for Prediction of Osteoporosis from Clinical Health Examination Data”. We have studied your comments carefully and made corresponding revision which we hope to meet your suggestions. The response to your questions or comments were shown in details in the following texts:
Point 1: L35 Please cite source reference, as [5] is about US population.
Response 1: Thank you for the comment. We have corrected the cited reference in the manuscript to “Chen FP, Huang TS, Fu TS, Sun CC, Chao AS, Tsai TL. Secular trends in incidence of osteoporosis in Taiwan: A nationwide population-based study. Biomed J. 2018;41(5):314-20”. This change was revised in the manuscript on Lines 34-35. (The prevalence of osteoporosis and osteoporotic fractures among Taiwanese more than 50 years old increased from 17.4% in 2001 to 25.0% in 2011(1).)
Point 2: L39 Since 2017 proper abbreviation dual-energy x-ray absorptiometry is DXA not DEXA. Please correct throughout the manuscript.
Response 2: Thank you for the suggestion. We have corrected the abbreviation throughout the manuscript.
Point 3: L42 Correct “bone marrow density” to “bone mineral density”
Response 3: Thank you for the correction. We have corrected “bone marrow density” to “bone mineral density” on Lines 41-43. (Currently, bone mineral density (BMD) examination is still not included as a generalized screening tool during regular health checkup in Taiwan.)
Point 4: L92 Suggestion – Is it possible to categorize these inputs into some groups?
Response 4: Thank you for the suggestion. During data procession, we have categorized the candidate features into different domains, as presented in the table below.
|
|
Domain of features |
Considered feature |
Selected feature |
|
History taking |
Personal history |
History of smoking History of alcohol drinking |
V V |
|
Medical history |
History of Hypertension History of Diabetes mellitus |
|
|
|
OBGYN history (for Female) |
Gravidity Parity Menopause status History of HRT |
V V V |
|
|
Physical examination |
Physical characteristics |
Age Height Weight Body fat Waist circumference Buttock circumference |
V V V
V
|
|
Vital signs |
Systolic blood pressure Diastolic blood pressure Pulse rate |
|
|
|
Laboratory test |
Hematological profile |
White blood count Hemoglobin Platelet |
V
|
|
Renal function |
Blood urea nitrogen Creatinine |
V |
|
|
Electrolytes |
Sodium Potassium Calcium Phosphate |
|
|
|
Liver function |
Total bilirubin Alanine transaminase Aspartate aminotransferase Alkaline phosphatase |
V
V |
|
|
Thyroid function |
Thyroid-stimulating hormone Free thyroxine |
V
|
|
|
Lipid profile |
Total cholesterol Triglyceride HDL-C LDL-C |
V V
|
|
|
Protein content |
Total protein Albumin |
V |
|
|
Markers for diabetes mellitus |
Fasting glucose Postprandial glucose Hemoglobin A1c |
|
|
|
Integrated results |
Hypertension Diabetes mellitus |
V V |
|
We have presented the summary in Supplementary Table S1.
Point 5: L99 Please provide some details about this "careful examination”. What was the procedure? Who performed selection? Single person or some council? Physician? If yes, of what specialization?
Response 5: Thank you for the suggestion. From the candidate features in the table above, we have tested the representativeness, availability, and significance on the prediction of decreased bone density. We aimed to select 1-2 features from each domain which were more representative, more commonly used, and showed significant difference between the normal BMD group and the decreased BMD group. This selection process was done by discussion among the authors, which consisted of doctors from the Department of Family Medicine and Department of Medical Education. We have clarify this process and revised our manuscript on Lines 101-108. (Features from these domains were examined for the representativeness, availability, and the significance on the prediction of decreased bone density. We selected 1-2 features from each domain which were more representative, more commonly used, and showed significant difference between the normal BMD group and the decreased BMD group. This selection process was done by discussion among the authors. After examination, 16 input features for men and 19 input features for women were selected.)
Point 6: L127 What do you mean be “rescaled”? Normalized?
Response 6: Thank you for the comment. “Normalization” and “rescaling data to [0, 1]” mean the same process. In the training dataset, we rescaled the data to [0, 1], which was the same as to normalize the data. This rescale index was recorded, and was used in rescaling the testing dataset. Which means that, the range of testing dataset after rescaling would not be [0, 1] unless the largest and smallest value were the same in the testing and training dataset.
Point 7: L154 I suggest adding the table in which all machine models are characterized - this will allow for a clearer characterization of the models used and to will help to visualize the differences between models used for men and women datasets.
Response 7: Thank you for the delicate suggestion. For hyperparameter tuning, different combination of the number of layers (1 or 2 hidden layers), number of nodes (from 4 to 20 in each hidden layer), learning rate (from 0.01 to 0.0001), and dropout rate (from 0% to 60%) were tested for the ANN model. Grid search was done for the SVM, RF, and KNN model. For the SVM model, the kernel type (linear, polynomial, or radial basis function), regularization parameter C (from 2-2 to 29), kernel coefficient gamma (from 2-9 to 22), and degree (from 1 to 4) for the polynomial kernel function were examined; for the RF model, the number of trees (from 100 to 1000), number of features to consider (from 3 to 11), and the maximum depth of the tree (from 3 to 11) were examined; for the KNN model, the number of neighbors (from 1 to 30), the leaf size (from 1 to 49), and the power parameter p (Manhattan distance or Euclidean distance) were tested. All models were constructed with a balanced class weight. For each set of hyperparameter, randomization for the split of validation dataset and the training process were repeated several times. Area under the receiver operating char-acteristic curve (AUROC) was calculated for the validation dataset in each training process, and the mean value of AUROC were compared. We adopted the hyperparameter that yield a better mean AUROC during validation to train our models. We have added this message in our manuscript on Lines 154-169, and presented in Table 1-1 to Table 1-4. (For hyperparameter tuning, different combination of the number of layers (1 or 2 hidden layers), number of nodes (from 4 to 20 in each hidden layer), learning rate (from 0.01 to 0.0001), and dropout rate (from 0% to 60%) were tested for the ANN model. Grid search was done for the SVM, RF, and KNN model. For the SVM model, the kernel type (linear, polynomial, or radial basis function), regularization parameter C (from 2-2 to 29), kernel coefficient gamma (from 2-9 to 22), and degree (from 1 to 4) for the polynomial kernel function were examined; for the RF model, the number of trees (from 100 to 1000), number of features to consider (from 3 to 11), and the maximum depth of the tree (from 3 to 11) were examined; for the KNN model, the number of neighbors (from 1 to 30), the leaf size (from 1 to 49), and the power parameter p (Manhattan distance or Euclidean distance) were tested. All models were constructed with a balanced class weight. For each set of hyperparameter, randomization for the split of validation dataset and the training process were repeated several times. Area under the receiver operating characteristic curve (AUROC) was calculated for the validation dataset in each training process, and the mean value of AUROC were com-pared. We adopted the hyperparameter that yield a better mean AUROC during validation to train our models.)
|
Table 1-1 Selection of hyperparameters for the ANN model. |
|
|
||
|
|
Number of hidden layers |
Number of nodes |
Learning rate |
Dropout rate |
|
Examine range |
1-2 |
4-20 in each hidden layer |
0.01-0.0001 |
0-60% |
|
Selected hyperparameters for the men model |
2 |
9 in hidden layer 1, 4 in hidden layer 2 |
0.001 |
40% |
|
Selected hyperparameters for the women model |
2 |
13 in hidden layer 1, 7 in hidden layer 2 |
0.001 |
40% |
|
ANN: Artificial neural network |
||||
|
Table 1-2 Selection of hyperparameters for the SVM model. |
|
|
||
|
|
Kernal type |
Regularization parameter C |
Kernel coefficient gamma |
Degree for polynomial function |
|
Grid search range |
linear, polynomial, or radial basis function |
2-2-29 |
2-9-22 |
1-4 |
|
Selected hyperparameters for the men model |
radial basis function |
25 |
2-3 |
- |
|
Selected hyperparameters for the women model |
radial basis function |
24 |
2-1 |
- |
|
SVM: Support vector machine |
||||
|
Table 1-3 Selection of hyperparameters for the RF model. |
|
||
|
|
Number of trees |
Number of features to consider |
Maximum depth of the tree |
|
Grid search range |
100-1000 |
3-11 |
3-11 |
|
Selected hyperparameters for the men model |
300 |
8 |
8 |
|
Selected hyperparameters for the women model |
600 |
10 |
10 |
|
RF: Random forest |
|||
|
Table 1-4 Selection of hyperparameters for the KNN model. |
|
||
|
|
Number of neighbors |
Leaf size |
Power parameter p |
|
Grid search range |
1-30 |
1-49 |
Manhattan distance or Euclidean distance |
|
Selected hyperparameters for the men model |
28 |
15 |
Manhattan distance |
|
Selected hyperparameters for the women model |
13 |
3 |
Euclidean distance |
|
KNN: K-nearest neighbors |
|||
Point 8: L174 ROC abbreviation was nod defined earlier (only AUROC, L151)
Response 8: Thank you for the suggestion. We have corrected the abbreviation on Lines 195-197. (When we tested these probabilities with the true condition of having osteoporosis or not, receiver operating characteristic (ROC) curves could be drawn.)
Point 9: Tables: the information which tests were used to calculate p-values should be given in each table footnote.
Response 9: Thank you for the suggestion. We have revised the tables and included the tests performed in the footnote in Table 2, Table 3, and Table 4.
Point 10: Table 1: give the p-values for normal and osteopenic bone densities
Response 10: Thank you for the suggestion. We performed Chi-square test for the categorical variable. We have revised Table 2, moved the position of p-value, and added explanation in the footnote to avoid misunderstanding.
|
Table 2. Baseline characteristics of the study population. |
|
|
||
|
|
Total (n=5982) |
Men (n=3053) |
Women (n=2929) |
p-value* |
|
Age (years) |
59.3 ± 7.0 |
59.3 ± 7.0 |
59.3 ± 7.0 |
0.9650 |
|
Body height (cm) |
161.8 ± 8.4 |
167.7 ± 6.2 |
155.7 ± 5.5 |
<0.0001 |
|
Body weight (kg) |
63.5 ± 11.7 |
69.8 ± 10.2 |
56.9 ± 9.2 |
<0.0001 |
|
Waist circumference (cm) |
83.9 ± 9.8 |
87.8 ± 8.4 |
79.9 ± 9.5 |
<0.0001 |
|
History of smoking (n, %) |
1255 (21.0) |
1133 (37.1) |
122 (4.2) |
<0.0001 |
|
History of alcohol drinking (n, %) |
484 (8.1) |
414 (13.6) |
70 (2.4) |
<0.0001 |
|
Diabetes mellitus (n, %) |
1075 (18.0) |
618 (20.2) |
457 (15.6) |
<0.0001 |
|
Hypertension (n, %) |
2226 (37.2) |
1250 (40.9) |
976 (33.3) |
<0.0001 |
|
Albumin (g/dL) |
4.50 ± 0.30 |
4.52 ± 0.27 |
4.45 ± 0.26 |
<0.0001 |
|
Hemoglobin (g/dL) |
14.1 ± 1.4 |
15.0 ± 1.2 |
13.2 ± 1.1 |
<0.0001 |
|
ALT (IU/L) |
27.5 ± 18.6 |
30.5 ± 20.0 |
24.4 ± 16.6 |
<0.0001 |
|
Creatinine (mg/dL) |
0.89 ± 0.27 |
1.03 ± 0.26 |
0.75 ± 0.21 |
<0.0001 |
|
TG (mg/dL) |
133.1 ± 84.8 |
147.2 ± 95.2 |
118.4 ± 69.1 |
<0.0001 |
|
HDL-C (mg/dL) |
55.1 ± 16.4 |
48.76 ± 13.54 |
61.78 ± 1.57 |
<0.0001 |
|
ALK-P (IU/L) |
67.9 ± 19.9 |
66.0 ± 18.4 |
69.9 ± 21.2 |
<0.0001 |
|
TSH (uIU/mL) |
2.32 ± 2.34 |
2.18 ± 2.15 |
2.47 ± 2.52 |
<0.0001 |
|
Menopause (n, %) |
2448 (83.6) |
|||
|
History of HRT (n, %) |
283 (9.7) |
|||
|
Parity (n) |
2.4 ± 1.4 |
|||
|
Categories of bone density result |
<0.0001 |
|||
|
Normal (n, %) |
3058 (51.1) |
1802 (59.0) |
1256 (42.9) |
|
|
Osteopenia (n, %) |
2503 (41.8) |
1134 (37.1) |
1369 (46.7) |
|
|
Osteoprosis (n, %) |
421 (7.0) |
117 (3.8) |
304 (10.4) |
|
|
* P-values were calculated with two-tailed T tests for continuous variables, two-tailed Z tests for binary variables, and Chi-square tests for categorical variables. ALT: alanine transaminase; TG: triglyceride; HDL-C: high-density lipoprotein cholesterol; ALK-P: alkaline phosphatase; TSH: thyroid-stimulating hormone; HRT: hormone-replacement therapy. |
||||
Point 11: L232-236 Repetition of the results - please remove
Response 11: Thank you for the suggestion. We have removed the footnote in Figure 2-2.
Point 12: L237-239 Is sound as final conclusion rather than one of the first sentences of the discussion.
Response 10: Thank you for the suggestion. We have modified the first paragraph of our discussion on Lines 255-262. (In this study, we used five different machine learning algorithms as ANN, SVM, RF, KNN, and LoR for the screening of osteoporosis in community-dwelling individu-als older than fifty years old. The models reached AUROC of 0.821-0.843 in men and 0.767-0.811 in women. At designated cutoff points, the models achieved 83-96% sensi-tivity and 53-73% specificity on prediction of osteoporosis in men; and 76-90% sensi-tivity and 62-69% specificity on prediction of osteoporosis in women. The ANN, SVM, RF, and LoR models in men, and the ANN, SVM, and RF models in women performed significantly better than the well-established OSTA model.)
Point 13: L293 Which machine learning model seems to be the most promising tool? Is there a winner?
Response 13: Thank you for the comment. In both gender’s models, the ANN, SVM, and RF models seemed to yield better results than other models. Of these, the RF model gained slightly better AUROC. However, it would be difficult to choose a definite winner due to the similarity in performance; also, in different population and different situation the models applied, the desired model and cut-off point could change accordingly. We considered that the models constructed in the study had similar performance. Which model to use, and which cut-off value to adopt would need to be decided case by case. We have added this content in our Conclusion on Lines 344-345. (It would be difficult to choose a definite winner of the models due to similarity in performance and application in different population and different situations.)
- Chen FP, Huang TS, Fu TS, Sun CC, Chao AS, Tsai TL. Secular trends in incidence of osteoporosis in Taiwan: A nationwide population-based study. Biomed J. 2018;41(5):314-20.

Round 2
Reviewer 1 Report
Thanks to the authors for their diligent attention to the comments made. For my part, I believe that the article has been substantially improved, so that it can be published in its present form.
Reviewer 2 Report
The paper Development of Machine Learning Models for Prediction of Osteoporosis from Clinical Health Examination Data can be published in the present form